# Learning from Sample Stability for Deep Clustering

Zhixin Li [1]   Yuheng Jia [1 2]   Hui Liu [3]   Junhui Hou [4]

## Abstract

Deep clustering, an unsupervised technique independent of labels, necessitates tailored supervision for model training. Prior methods explore supervision like similarity and pseudo labels, yet overlook individual sample training analysis. Our study correlates sample stability during unsupervised training with clustering accuracy and network memorization on a per-sample basis. Unstable representations across epochs often lead to mispredictions, indicating difficulty in memorization and atypicality. Leveraging these findings, we introduce supervision signals for the first time based on sample stability at the representation level. Our proposed strategy serves as a versatile tool to enhance various deep clustering techniques. Experiments across benchmark datasets showcase that incorporating sample stability into training can improve the performance of deep clustering. The code is available at https://github.com/LZX-001/LFSS.

## 1. Introduction

Deep clustering trains a deep neural network in an unsupervised manner to assign appropriate labels to samples. Unlike supervised learning, where the ground truth labels are used to guide model optimization, deep clustering lacks such labels to aid network training. Consequently, suitable supervision signals must be derived from the data itself to enable deep clustering models to extract meaningful features, yield discriminative high-dimensional representations, and achieve accurate clustering (Lu et al., 2024).

---
[1]School of Computer Science and Engineering, Southeast University, Nanjing 210096, China [2]Key Laboratory of New Generation Artificial Intelligence Technology and Its Interdisciplinary Applications (Southeast University), Ministry of Education, China [3]School of Computing Information Sciences, Saint Francis University, Hong Kong, China [4]Department of Computer Science, City University of Hong Kong, Hong Kong, China. Correspondence to: Yuheng Jia <yhjia@seu.edu.cn>.

*Proceedings of the $42^{nd}$ International Conference on Machine Learning*, Vancouver, Canada. PMLR 267, 2025. Copyright 2025 by the author(s).

Prior studies have utilized various supervision signals such as autoencoder reconstruction (Xie et al., 2016; Li et al., 2018; Yang et al., 2017), sample similarity and neighbor relationship (Chang et al., 2017; Ji et al., 2019; Tao et al., 2021; Li et al., 2024; Huang et al., 2019; 2023), and pseudo labels (Van Gansbeke et al., 2020; Niu et al., 2022; Tian et al., 2017; Cai et al., 2023). Recent advancements in self-supervised learning (Grill et al., 2020; He et al., 2020; Chen et al., 2020) have prompted many deep clustering methods to explore contrastive learning for generating clustering-compatible representations (Huang et al., 2023; Li et al., 2022; 2021b; Yu et al., 2023). Despite the effectiveness of these supervision signals in enhancing clustering performance, current research often overlooks the learning progress of individual samples during network training.

Throughout training, the representations of the same input sample may vary across epochs due to ongoing weight updates. We propose a new metric called **sample stability**, which is defined the cosine similarity of representations for the same input across two consecutive epochs in the model, to represent the learning progress of individual samples. Through extensive experiments, we consistently observe that samples with the most unstable representations are prone to incorrect clustering (see **Observation 1**). In addition, we find that unstable samples exhibit non-random long-term instability during training (see **Observation 2**). Further examination reveals that consistently unstable samples are typically atypical and rare compared to stable counterparts, indicating their difficulty in model memorization (see **Observation 3**). These observations hold true across multiple self-supervised learning and deep clustering methods, as well as across various datasets and epochs, detailed in Section 2. The above insights underscore the utility of sample stability in evaluating individual sample training progress. Thus, we pioneer the use of sample stability to construct supervision signals and introduce a novel deep clustering method LFSS (Learning From Sample Stability). Our approach incorporates a predecessor network to track previous epoch weights and assess sample stability. Moreover, based on the contrastive learning paradigm (Chen et al., 2020), we employ sample stability from both the instance level and the cluster level.

In summary, this work makes the following key contributions:

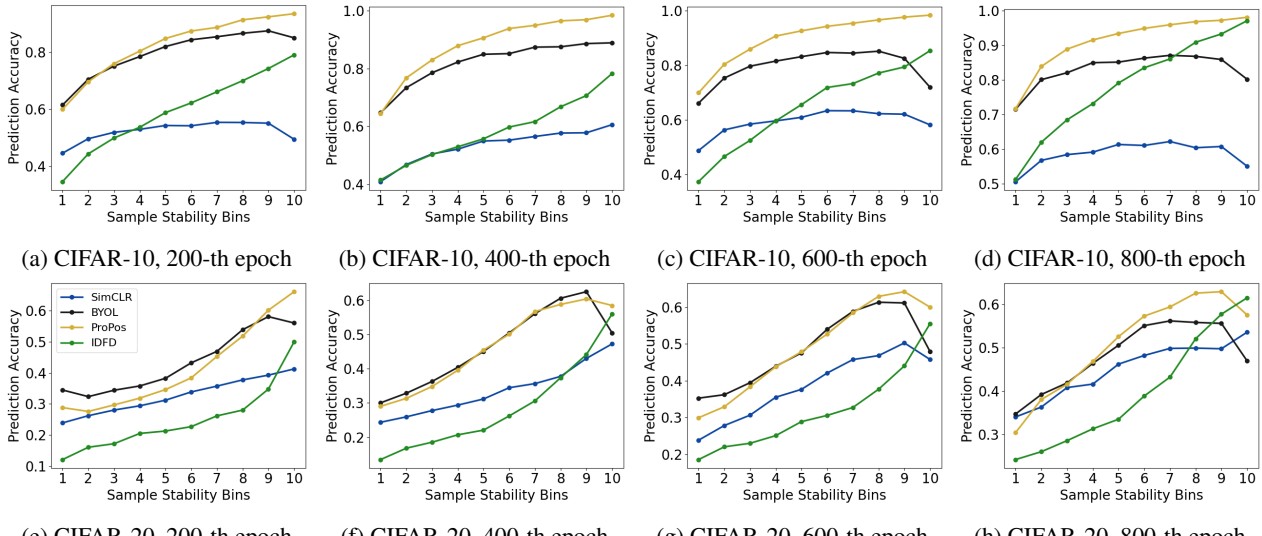

*Figure 1.* Connection between sample stability and clustering prediction accuracy on CIFAR-10 and CIFAR-20. Four methods under various epochs: SimCLR (blue), BYOL (black), ProPos (yellow), IDFD (green) are shown. It is clear that unstable samples are more likely to be predicted incorrectly. All subfigures share the same legend.

- For the first time, we identify the strong correlation between sample stability, clustering prediction, and network memorization in the context of unsupervised model training. Notably, unstable samples are prone to erroneous predictions, and we establish a clear link between sample stability and network memorization. Our extensive experiments across various methods and datasets validate these findings and reinforce our perspectives.

- We pioneer the utilization of sample stability at the representation level as a guiding signal in deep clustering. While recent approaches have introduced diverse supervision signals to enhance deep clustering, they often overlook the evolution of individual samples during training. Recognizing that sample stability can effectively reflect training quality in an unsupervised setting, we harness it as a guiding signal, introducing a novel deep clustering methodology at both the instance and cluster levels.

- We assess the clustering performance of LFSS across multiple widely utilized datasets and benchmark it against state-of-the-art techniques. Our experimental results unequivocally showcase the efficacy of our approach. Furthermore, we illustrate that our method can be seamlessly integrated to enhance the performance of existing approaches, serving as a valuable plugin.

## 2. Our Insights

In this section, we delve into the common phenomena across different unsupervised deep clustering methods. Based on

the observations from substantial experiments, we elucidate the relationships of **sample stability** with both **clustering prediction** and **network memorization** for the first time. In unsupervised learning, labels are not accessible, and accordingly, we cannot use the accuracy of label prediction to evaluate the training status of a single sample. To this end, we exploit the high-dimensional representations of samples in the embedding space, and use the variations at the representation level as the criterion for assessing the training status of a sample. Specifically, we define sample stability of a sample as the cosine similarity of the representations of the same sample across two consecutive epochs. Assume $f(\cdot)$ is the deep neural network, and $x$ is an input sample, then sample stability in the $t$-th epoch can be represented as follow:

$$s(f, x, t) = \frac{(f^t(x))^T f^{t-1}(x)}{\|f^t(x)\|\|f^{t-1}(x)\|}, \quad (1)$$

where $f^t(\cdot)$ denotes the network at the $t$-th epoch, and $\|\cdot\|$ denotes $\ell_2$ norm. In other words, lower sample stability indicates that the representation of a sample is unstable during specific epochs in the training process. Based on the above definition, we have the following three observations.

**Observation 1: Unstable samples are more likely to be predicted incorrectly.** We observe that sample stability is closely associated with clustering prediction. To validate it, we train several representative unsupervised learning models: SimCLR (Chen et al., 2020) and BYOL (Grill et al., 2020) are two prevalent contrastive learning paradigms and the cornerstones of recent deep clustering methods; ProPos (Huang et al., 2023) and IDFD (Tao et al., 2021) are representative and state-of-the-art deep clustering methods in

recent years. We adopt ResNets (He et al., 2016) as the backbone. We adopt K-means (Hartigan & Wong, 1979) to cluster as it is the most commonly employed clustering algorithm in deep clustering (Huang et al., 2023; Tao et al., 2021; Li et al., 2023; Niu et al., 2022). We train the models and record the sample stability at different epochs. We divide all samples into 10 bins based on their stability, from smallest to largest. For statistical fairness, the number of samples in each bin should be similar to avoid outliers. So each bin has 10% of the samples. We present the results on the CIFAR-10 and CIFAR-20 (Krizhevsky, 2009) in Figure 1. Experiments on ImageNet-10 (Chang et al., 2017), Tiny-ImageNet (Le & Yang, 2015) and ImageNet-1K (Deng et al., 2009) are presented in Appendix A. We can observe that under all conditions (across different datasets, methods, and epochs), unstable samples are more likely to be predicted incorrectly than stable samples. This suggests that unstable samples are often poorly learned and difficult to be grouped into a specific class. More observations and discussions can be found in Appendix A.

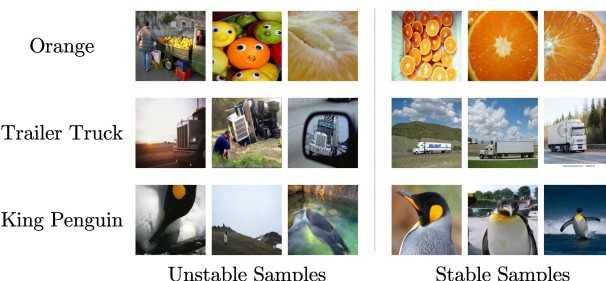

*Figure 2.* Visualization results of SimCLR on stable and unstable samples from ImageNet-10. We can observe that unstable samples tend to be visually more complex or unusual. Such samples can be considered atypical or rare, making them hard to memorize, while the stable samples are often typical and conventional.

*Table 1.* Recurrence counts of unstable samples on CIFAR-10. Some samples exhibit long-term instability.

| Count | SimCLR | BYOL | ProPos | IDFD | Random |
|---|---|---|---|---|---|
| Count = 1 | 11169 | 10325 | 8593 | 10159 | 17496 |
| Count = 2 | 3834 | 3771 | 3791 | 3981 | 2916 |
| Count = 3 | 1377 | 1499 | 1774 | 1517 | 216 |
| Count = 4 | 258 | 409 | 626 | 332 | 6 |

**Observation 2: Some samples exhibit long-term instability in the training process.** To verify whether the difficulty in accurately predicting unstable samples is due to the samples themselves rather than the networks, we conduct additional experiments by tracking unstable samples over multiple epochs. Specifically, we count the recurrence of the top 10% unstable samples at the 200-th, 400-th, 600-th, and 800-th epochs on CIFAR-10. As a reference, we also randomly assign 10% samples as the most unstable samples at each epoch and count their recurrences. As shown in Table 1, we observe that some unstable samples reappear repeatedly at different epochs. Moreover, the numbers of samples experiencing instability only once for deep clustering methods are observed to be lower than what would be expected under random conditions. Conversely, the numbers of samples manifesting multiple instabilities significantly exceed the expectations based on random chance. The above phenomena suggest that some unstable samples are persistently unstable (long-term instability) rather than being sporadically. Results on other datasets are shown in Appendix B.

**Observation 3: Sample stability is closely associated with network memorization.** As shown in Table 1, some samples fail to learn stable representations during model

training. Such samples can be considered difficult for the network to memorize. We further investigate the characteristics of those unstable samples by training SimCLR (Chen et al., 2020) on the ImageNet-10 dataset. Specifically, we track the top 10% stable and the top 10% unstable samples at the 200-th, 400-th, 600-th, and 800-th training epochs. We select samples that consistently appear in three or four of these training epochs and consider them to be the representative examples of long-term stable or unstable instances. As shown in Figure 2, we can observe that long-term stable samples are often typical and conventional, with complete presentations and distinctive features, while long-term unstable samples appear to be rare and atypical, such as, oddly decorated oranges, a trailer truck in a rearview mirror, a penguin in unusual lighting. Compared to stable samples that are easily recognizable by human eyes, unstable samples are visually more complex, with unusual perspectives, incomplete or abnormal compositions, or they may not be the main focus of the image. Such rare or atypical samples are difficult for the network to memorize due to their lack of representativeness.

## 3. Related Work

### 3.1. Contrastive Learning

Contrastive learning is an effective self-supervised learning approach (Gui et al., 2024; Wu et al., 2018). Recently, many well-known contrastive learning approaches were proposed (Grill et al., 2020; He et al., 2020; Chen et al., 2020; Caron et al., 2020; Chen & He, 2021; Zbontar et al., 2021; Bardes et al., 2022). Among them, the methods based on negative sample pairs have gained widespread influence. Specifically, each sample undergoes data augmentation to produce different views. The views from the same sample are regarded as positive samples, while others are negative samples. By minimizing the distance between positive samples and max-

imizing the distance between negative samples in the latent space, effective representations are learned. Considering NT-Xent loss (Normalized Temperature-scaled Cross Entropy Loss) in SimCLR (Chen et al., 2020), in a dataset of size $N$, each sample is augmented twice and this results in $2N$ views. Let $z_i$ be the representation of the $i$-th view. The NT-Xent loss can be formulated as follows:

$$L_{\text{NT-Xent}} = -\frac{1}{2N} \sum_{i=1}^{2N} \log \frac{\exp(\text{sim}(z_i, z_i^+)/\tau)}{\sum_{k=1}^{2N} \mathbf{1}_{[k \neq i]} \exp(\text{sim}(z_i, z_k)/\tau)}, \quad (2)$$

where $\text{sim}(u,v) = u^T v / \|u\|\|v\|$ is the cosine similarity of representation $u$ and $v$, $\tau$ is a temperature parameter and $\mathbf{1}_{[k \neq i]}$ returns 1 when $k \neq i$ else 0. $z_i^+$ is the positive sample of $z_i$. Eq. (2) serves to pull positive samples closer while pushing negative samples further apart, thereby learning appropriate representations.

Apart from contrastive learning methods that require negative samples, some self-distillation methods can effectively train models using only positive samples (Grill et al., 2020; Chen & He, 2021). These methods often contrast representations produced by asymmetric networks. Considering $f_t$ is a target network, $f_o$ is an online network and $h$ is a prediction head, the loss in a self-distillation method is typically constructed as:

$$L_{\text{sd}} = \frac{1}{N} \sum_{i=1}^{N} 2 - 2 \cdot \text{sim}(f_t(x_i), h(f_o(x_i^+))), \quad (3)$$

where $x_i$ and $x_i^+$ are augmented versions of the $i$-th sample.

### 3.2. Deep Clustering

Deep clustering aims to train deep neural networks in the absence of labels to perform clustering tasks. Compared to supervised learning, it can significantly save time and labor in label annotation. Although labels are not required, supervision signals are still needed to provide feedback for model training. Therefore, various supervision signals have been proposed to guide model training (Xie et al., 2016; Li et al., 2018; Yang et al., 2017; Chang et al., 2017; Ji et al., 2019; Tao et al., 2021; Jia et al., 2021; Peng et al., 2022; Li et al., 2024; Liu et al., 2024; Jia et al., 2025). Specifically, some pseudo-labeling approaches leverage prediction consistency across different epochs or models to filter unreliable training samples (Mahon & Lukasiewicz, 2023; 2021). However, the stability of samples at the representation level remains underexplored. In recent years, due to the advancements in contrastive learning (He et al., 2020; Grill et al., 2020; Chen et al., 2020), contrastive learning has been extensively adopted in deep clustering, leading to excellent clustering performance (Huang et al., 2023; Li et al., 2022; 2021b; Yu et al., 2023; Qi et al., 2024; Li & Jia, 2025).

### 3.3. Network Memorization

A deep neural network (DNN) is a universal approximator that can represent any function when its capacity is sufficiently large (Hornik et al., 1989; Cybenko, 1989). Some researchers study the learning mechanisms in networks, particularly how networks memorize samples, especially atypical ones, in the training progress (Maini et al., 2023). In supervised learning, previous researches (Maennel et al., 2020; Arpit et al., 2017; Zhang et al., 2017; Maini et al., 2023) have revealed that DNNs can memorize samples with random labels. By comparing training with real labels and random labels, they explain how networks memorize. Also, previous works (Feldman, 2020; Feldman & Zhang, 2020) demonstrated that memorizing rare and atypical samples is crucial for achieving close-to-optimal generalization error in supervised classification tasks. In unsupervised learning, how to identify and memorize rare and atypical samples is worth studying. However, there is only a limited body of related works in this field (Hooker et al., 2019; Jiang et al., 2021), further explorations are needed.

## 4. Proposed Method

Based on the observations discussed in Section 2, we illustrate that the stability of samples can serve as an indicator of the training advancement of each sample. Samples that exhibit instability are prone to erroneous clustering and are challenging for the network to memorize. Consequently, we can utilize sample stability as a guiding signal to steer the training of models in deep clustering. In this paper, we propose a novel deep clustering methodology called LFSS (Learning From Sample Stability), which capitalizes on sample stability signals from both the instance and cluster levels, in conjunction with self-distillation.

As shown in Figure 3, we adopt an asymmetric network structure as in (Grill et al., 2020). $f_t$ and $f_o$ denote the target network and online network, respectively. $f_p$ is the proposed predecessor network for measuring sample stability. $h$ is an MLP predictor head. The stop-gradient operation is applied to both $f_t$ and $f_p$. $f_t$ is momentum updated from $f_o$ with $\theta_t = m\theta_t + (1 - m)\theta_o$, where $m \in [0, 1]$ is a target decay rate, $\theta_*$ denotes parameters in the network $f_*$ and the asterisk ($*$) is a wildcard character. $f_p$ is directly copied from $f_t$ in the last epoch. Given a dataset of size $N$:$\{x_i\}_{i=1}^N$, each sample $x_i$ is augmented twice to generate two views: $x_i^t$ and $x_i^o$, intended for input into $f_t$ and $f_o$ respectively. The overall loss of LFSS comprises three components: instance-level loss for LFSS $L_I$, cluster-level loss for LFSS $L_C$, and self-distillation with noise loss $L_S$.

**Instance-level loss for LFSS.** We construct an instance-level supervision signal with sample stability. As introduced in Observation 1, sample with unstable representations are

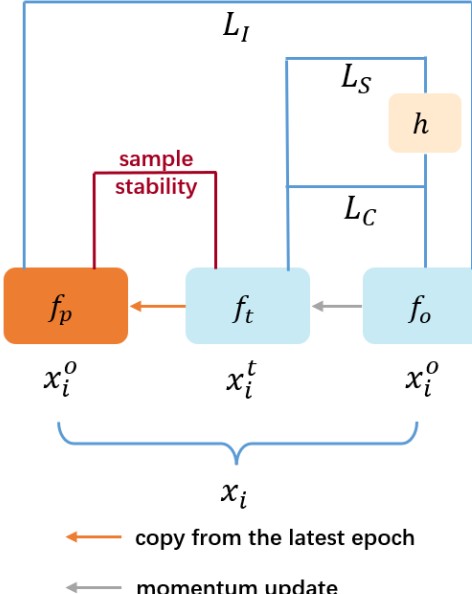

*Figure 3.* The network architecture of LFSS. Each input is augmented into two views ($x_i^o$ and $x_i^t$), which are then fed into the networks ($f_p$, $f_t$ and $f_o$) as inputs. Only the online network $f_o$ and the predictor head $h$ are involved in gradient backpropagation. The target network $f_t$ is momentum updated from $f_o$, and the predecessor network $f_p$ is copied from $f_t$ in the latest epoch and is used to measure sample stability. Based on the outputs from different networks, we measure the three components in the objective function, including instance-level loss $L_I$, cluster-level loss $L_C$, and self-distillation with noise loss $L_S$.

likely to be clustered incorrectly. However, forcing the representations from two consecutive epochs to be as close as possible could potentially impede the training progress. To balance stability and model training, our approach aims to penalize excessive variations of representations while ensuring that these penalties do not harm the overall evolution and improvement of the model. Therefore, we extend the NT-Xent loss and formulate the instance-level LFSS loss as:

$$L_I = -\frac{1}{N} \sum_{i=1}^{N} \log \frac{\exp(\mathrm{sim}(f_o(x_i^o), f_p(x_i^o))/\tau)}{\sum_{k=1}^{N} \mathbf{1}_{[k \neq i]} \exp(\mathrm{sim}(f_o(x_i^o), f_p(x_k^o))/\tau)}. \quad (4)$$

In Eq. (4), the output representations of the same sample in two consecutive epochs: $f_o(x_i^o)$ and $f_p(x_i^o)$, form positive sample pairs, while those from different samples form negative sample pairs. The positive samples output by the predecessor network result in larger losses for unstable samples, whereas negative samples prevent the network from staying the same and benefit model training.

**Cluster-level loss for LFSS.** In clustering, cluster centers can describe the main characteristics of their corresponding clusters. Discriminative cluster centers help generate better

sample partitioning. To this end, we apply contrastive learning on the cluster level to obtain more discriminative cluster centers. We assume that the cluster number $K$ is known as a priori, and adopt K-means (Hartigan & Wong, 1979) to obtain the clustering partitions. The center for the $i$-th cluster can be obtained as follow:

$$c_i^* = \frac{\sum_{j \in C_i} f_*(x_j^*)}{\| \sum_{j \in C_i} f_*(x_j^*) \|}, \quad (5)$$

where $c_i^*$ is the center of $i$-th cluster, obtained from the representations output by network $f_*$, and $C_i$ is the set of sample indices belonging to the $i$-th cluster. Note that we perform clustering on the original samples instead of augmented ones to form $\{C_i\}_{i=1}^K$.

As unstable samples are more likely to be grouped in the incorrect clusters, excluding them can lead to more accurate cluster centers. Thus, we exclude the most unstable instances in the dataset with an unstable ratio $\delta \in [0, 1]$. $N_s = \lfloor N * (1 - \delta) \rfloor$ indicates the number of remaining stable samples and the set $S$ contains indices of these remaining stable samples:

$$S = \mathrm{argsort}(\mathrm{sim}(f_t(x_i), f_p(x_i)))[-N_s :], \quad (6)$$

where $\mathrm{argsort}(\cdot)$ is a function that returns an array representing the indices of the input elements sorted in ascending order. $[-N_s :]$ returns the last $N_s$ elements in the array. In this way, we record the stable samples in the set $S$. Excluding the most unstable samples, we can obtain more representative cluster centers in the below protocol:

$$c_i^* = \frac{\sum_{j \in C_i \cup S} f_*(x_j^*)}{\| \sum_{j \in C_i \cup S} f_*(x_j^*) \|}. \quad (7)$$

Inspired by (Huang et al., 2023), we integrate cluster centers into contrastive learning. Specifically, we treat centers from the same clusters as positive pairs and those from different clusters as negative pairs:

$$L_C = -\frac{1}{K} \sum_{i=1}^{K} \log \frac{\exp(\mathrm{sim}(c_i^o, c_i^t))/\tau)}{\sum_{k=1}^{K} \mathbf{1}_{[k \neq i]} \exp(\mathrm{sim}(c_i^o, c_k^t)/\tau)}. \quad (8)$$

In each epoch, we first conduct K-means (Hartigan & Wong, 1979) globally for cluster assignments and then select stable samples in Eq. (6). The cluster centers in Eq. (8) are computed in a batch-wise manner.

**Self-distillation with noise.** In addition to the aforementioned methods that leverage sample stability, inspired by (Kingma & Welling, 2014; Huang et al., 2023), we adopt a self-distillation approach to learn robust representations by exploiting the noise. Specifically, we add random noise to the output of the online network $f_o$, which serves as the input to the predictor head $h$. The method is implemented

---

**Algorithm 1** Proposed LFSS

---

**Input:** Dataset $\mathbf{X}$ of size $N$, epoch number $T$, batch size $b$, cluster num $K$, hyper-parameter $\lambda$, $\eta$, $\sigma$ and $\delta$, online network $f_o$, target network $f_t$, predictor head $h$, predecessor network $f_p$.
**Output:** Clusters $\{C_i\}_{i=1}^K$
**for** $t = 1$ **to** $T$ **do**
    sample a batch $\{x_i\}_{i=1}^b$ from $\mathbf{X}$
    augment $x_i$ to generate $x_i^t$ and $x_i^o$
    compute self-distillation with noise $L_S$ using Eq. (9)
    compute instance-level loss for LFSS $L_I$ using Eq. (4)
    **if** $i > \eta$ **then**
        compute sample stability and select stable samples using Eq. (6)
        compute cluster centers using Eq. (7)
        compute cluster-level loss for LFSS $L_C$ using Eq. (8)
    **end if**
    update $f_o$ and $h$ by minimizing Eq. (10)
    update $f_p$ by directly copying $f_t$
    update $f_t$ using momentum update
**end for**
Apply K-means to obtain Clusters $\{C_i\}_{i=1}^K$

---

as below:

$$L_S = \frac{1}{N} \sum_{i=1}^N 2 - 2 \cdot \text{sim}(f_t(x_i^t), h(f_o(x_i^o) + \sigma\xi)), \quad (9)$$

where $\sigma > 0$ is a hyper-parameter indicating the intensity of noise and $\xi$ is a noise vector whose elements are drawn independently from the standard normal distribution $\mathcal{N}(0,1)$. By minimizing Eq. (9), we can learn more robust representations, which is beneficial for clustering.

**Model Training and Clustering Prediction.** The overall objective function for LFSS is

$$L = L_S + \lambda(L_I + L_C), \quad (10)$$

where $L_S$, $L_I$ and $L_C$ are losses in self-distillation with noise, instance-level LFSS and cluster-level LFSS, respectively. we set $\lambda$ as a trade-off hyper-parameter to balance two counterparts. To obtain suitable cluster centers, we will first train the neural network for $\eta$ epochs using only $L_S$ and $L_I$ as a warm-up stage, and then use the complete loss in Eq. (10) for the remaining epochs. After the model training is completed, we apply K-means (Hartigan & Wong, 1979) to obtain the clustering results. The overall procedure is summarized in Algorithm 1.

# 5. Experiments

In this section, we evaluate the proposed LFSS by comparing it with 15 state-of-the-art deep clustering approaches

on five commonly used datasets, and two more challenging large-scale datasets.

## 5.1. Datasets

We conduct experiments on multiple commonly used datasets, including CIFAR-10 (Krizhevsky, 2009), CIFAR-20 (Krizhevsky, 2009), STL-10 (Coates et al., 2011), ImageNet-10 (Chang et al., 2017), ImageNet-Dogs (Chang et al., 2017), Tiny-ImageNet (Le & Yang, 2015) and ImageNet-1K (Deng et al., 2009). CIFAR-20 is CIFAR-100 using 20 super-classes for evaluation. We train models on STL-10 with extra unlabeled data. ImageNet-10, ImageNet-Dogs and Tiny-ImageNet are subsets of ImageNet-1K, containing 10, 15, 200 classes respectively. A summary of the datasets used for evaluation is shown in Table 2.

*Table 2.* A summary of benchmark datasets used for evaluation.

| Dataset | #Samples | #Clusters | Image Size |
|---------|----------|-----------|------------|
| CIFAR-10 | 60,000 | 10 | 32×32×3 |
| CIFAR-20 | 60,000 | 20 | 32×32×3 |
| STL-10 | 13,000 | 10 | 96×96×3 |
| ImageNet-10 | 13,000 | 10 | 96×96×3 |
| ImageNet-Dogs | 19,500 | 15 | 96×96×3 |
| Tiny-ImageNet | 100,000 | 200 | 64×64×3 |
| ImageNet-1K | 1,281,167 | 1,000 | 224×224×3 |

## 5.2. Implementation Details

We adopt ResNet-18 as the backbone unless specifically specified. We train the models for 1,000 epochs with a batch size of 256, unless noted otherwise. We report the results of the last epoch, using three commonly used metrics: clustering accuracy (ACC), normalized mutual information (NMI), adjusted rand index (ARI). We adopt the stochastic gradient descent (SGD) optimizer and the cosine decay learning rate schedule to effectively train our model. Besides, we adopt data augmentation methods following (Chen et al., 2020). We empirically set the trade-off hyper-parameter $\lambda$ in Eq. (10) to 0.1 for all experiments unless otherwise specified. We set the unstable ratio $\delta$ to 0.1 for all experiments to exclude the most unstable samples in cluster-level loss for LFSS, as indicated by the results in Observation 1. We set the warmup epoch number $\eta$ to 200 for CIFAR-10 and ImageNet-10, 500 for CIFAR-20, STL-10 and ImageNet-Dogs. The noise intensity in Eq. (9) $\sigma$ is set to 0.01 for STL-10 and ImageNet-10, 0.001 for CIFAR-10, CIFAR-20 and ImageNet-Dogs. The analysis of these hyper-parameters is presented in Appendix E. We adopt K-means (Hartigan & Wong, 1979) to obtain clustering results and report the average results of 10 trials to avoid the impact of random initialization in K-means. All experiments are conducted based on PyTorch and all models are trained on an NVIDIA RTX 4090 GPU. Other training specifics can be found in

*Table 3.* Comparisons with various methods on clustering performance (in percent %) across five benchmarks datasets.

| Methods | CIFAR-10 | | | CIFAR-20 | | | STL-10 | | | ImageNet-10 | | | ImageNet-Dogs | | |
|---|---|---|---|---|---|---|---|---|---|---|---|---|---|---|---|
| | NMI | ACC | ARI | NMI | ACC | ARI | NMI | ACC | ARI | NMI | ACC | ARI | NMI | ACC | ARI |
| IIC | 51.3 | 61.7 | 41.1 | - | 25.7 | - | 43.1 | 49.9 | 29.5 | - | - | - | - | - | - |
| DCCM | 49.6 | 62.3 | 40.8 | 28.5 | 32.7 | 17.3 | 37.6 | 48.2 | 26.2 | 60.8 | 71.0 | 55.5 | 32.1 | 38.3 | 18.2 |
| PICA | 56.1 | 64.5 | 46.7 | 29.6 | 32.2 | 15.9 | - | - | - | 78.2 | 85.0 | 73.3 | 33.6 | 32.4 | 17.9 |
| SCAN | 79.7 | 88.3 | 77.2 | 48.6 | 50.7 | 33.3 | 69.8 | 80.9 | 64.6 | - | - | - | - | - | - |
| NMM | 74.8 | 84.3 | 70.9 | 48.4 | 47.7 | 31.6 | 69.4 | 80.8 | 65.0 | - | - | - | - | - | - |
| CC | 70.5 | 79.0 | 63.7 | 43.1 | 42.9 | 26.6 | 76.4 | 85.0 | 72.6 | 85.9 | 89.3 | 82.2 | 44.5 | 42.9 | 27.4 |
| MiCE | 73.7 | 83.5 | 69.8 | 43.6 | 44.0 | 28.0 | 63.5 | 75.2 | 57.5 | - | - | - | 42.3 | 43.9 | 28.6 |
| GCC | 76.4 | 85.6 | 72.8 | 47.2 | 47.2 | 30.5 | 68.4 | 78.8 | 63.1 | 84.2 | 90.1 | 82.2 | 49.0 | 52.6 | 36.2 |
| PCL | 80.2 | 87.4 | 76.6 | 52.8 | 52.6 | 36.3 | 71.8 | 81.0 | 67.0 | 84.1 | 90.7 | 82.2 | 44.0 | 41.2 | 29.9 |
| TCC | 79.0 | 90.6 | 73.3 | 47.9 | 49.1 | 31.2 | 73.2 | 81.4 | 68.9 | 84.8 | 89.7 | 82.5 | 55.4 | 59.5 | 41.7 |
| IDFD | 71.4 | 81.5 | 66.6 | 41.8 | 41.1 | 26.5 | 62.3 | 74.1 | 55.0 | 75.5 | 86.2 | 69.0 | 50.2 | 55.3 | 36.9 |
| ProPos | 85.6 | 92.0 | 84.1 | 54.9 | 51.8 | 37.8 | 72.1 | 83.2 | 70.4 | 81.2 | 89.0 | 78.7 | 54.8 | 63.9 | 45.2 |
| CoNR | 78.1 | 85.3 | 71.1 | 53.5 | 49.9 | 35.8 | 70.2 | 81.2 | 69.2 | 79.2 | 87.7 | 75.0 | 52.7 | 62.2 | 42.1 |
| DMICC | 72.2 | 83.1 | 67.5 | 40.0 | 41.8 | 24.3 | 64.1 | 75.5 | 60.5 | 84.3 | 92.5 | 85.2 | 55.1 | 54.2 | 39.8 |
| BYOL | 72.3 | 83.3 | 68.0 | 51.9 | 49.3 | 34.5 | 68.0 | 80.7 | 63.5 | 73.2 | 82.1 | 67.4 | 51.8 | 59.7 | 41.3 |
| LFSS (Ours) | **87.2** | **93.4** | **86.6** | **59.9** | **58.7** | **43.5** | **77.1** | **86.1** | **74.0** | **85.6** | **93.2** | **85.7** | **61.7** | **69.1** | **53.3** |

Appendix D.

## 5.3. Main Results

In this section, we evaluate the clustering performance of the proposed LFSS on five commonly used benchmark datasets and compare it with both baseline and state-of-the-art methods, as shown in Table 3. We compare LFSS with 14 representative deep clustering methods, including IIC (Ji et al., 2019), DCCM (Wu et al., 2019), PICA (Huang et al., 2020), SCAN (Van Gansbeke et al., 2020), NMM (Dang et al., 2021), CC(Li et al., 2021b), MiCE (Tsai et al., 2021), GCC (Zhong et al., 2021), PCL (Li et al., 2021a), TCC (Shen et al., 2021), IDFD (Tao et al., 2021), ProPos (Huang et al., 2023), CoNR (Yu et al., 2023), DMICC (Li et al., 2023) and baseline BYOL (Grill et al., 2020). The results from the IIC to PCL methods are directly quoted from (Huang et al., 2023). The remaining four methods, including IDFD, ProPos, CoNR, DMICC, are representative approaches recently. To ensure a fair comparison, we reproduced these methods using the same backbone, batch size, and number of epochs as our method. To ensure the validity of the results, consistent with our approach, we report the average outcomes of 10 K-means runs. Regarding other experimental setups of each method, we referred to their respective papers. We reproduced the baseline BYOL using exactly the same experimental setup as our method.

As listed in Table 3, it can be observed that the proposed LFSS outperforms the state-of-the-art methods on all datasets. Compared to baseline BYOL, LFSS has achieved significant improvements across all datasets. Specifically, it attains a significant improvement with substantial margins ranging from 5.4% to 18.6% in all metrics. Compared to state-of-the-art methods, our approach also achieves the

best results on all datasets, which demonstrates the effectiveness of the proposed LFSS. Specifically, on the fine-grained dataset ImageNet-Dogs, our method has achieved the best performance improvements, with increases of 5.2% in ACC, 6.3% in NMI, and 8.1% in ARI. We also conduct significance tests on the methods being implemented in Appendix F, demonstrating that our approach is statistically significantly better.

Additionally, we conduct experiments on large-scale datasets, such as Tiny-ImageNet and ImageNet-1K. LFSS achieves superior performance, demonstrating its effectiveness on large-scale datasets. These experimental results can be found in Appendix C.

Given the improvements of this method over other state-of-the-art methods and the baseline, we can attribute the performance improvements to the effective use of instance-level loss for LFSS and cluster-level loss for LFSS. We demonstrate that using sample stability as a supervision signal for deep clustering models can yield more discriminative representations and achieve better clustering performance.

## 5.4. Ablation Study

To evaluate the effectiveness of each component in LFSS, we have conducted extensive ablation studies on CIFAR-10 in Table 4. Note that all ablation experiments are conducted under the same experimental setup. We have demonstrated that our method achieves significant performance improvements compared to BYOL in Section 5.3. We further examine the effect of the self-distillation with noise method by training for 1,000 epochs using only the $L_S$ loss. Compared to BYOL, it achieves performance improvements, demonstrating that adding noise to the input of the predictor head

is effective. By separately removing $L_I$ and $L_C$ from the complete method, we find that compared with LFSS, both LFSS w/o $L_I$ and LFSS w/o $L_C$ face performance degradation, but there is still a performance improvement compared to the baseline. These results prove that $L_I$ and $L_C$ are beneficial for deep clustering. Additionally, we conduct experiments by removing the predecessor network. Specifically, we retain all loss functions of LFSS but no longer use the predecessor network (for generating sample stability) as a supervision signal. $L_I$ is computed by the output representations from $f_t$ and $f_o$, using different augmented inputs. $L_C$ is computed when unstable samples are not excluded. The results suggest that the modifications made using sample stability have good effects. The above ablation studies demonstrate that the effective utilization of sample stability leads to better clustering performance.

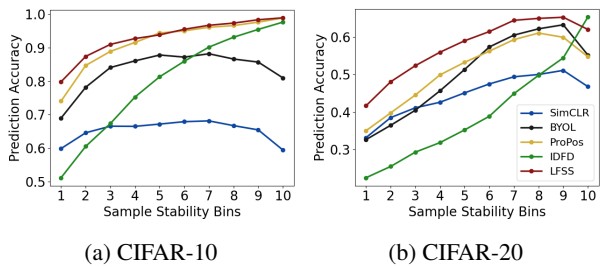

(a) CIFAR-10        (b) CIFAR-20

*Figure 4.* LFSS benefits the training of unstable samples, improving their clustering accuracy. The subfigures share the same legend.

*Table 5.* The proportion (in percent %) of samples transitioning from unstable to stable starting from the 200-th epoch.

|  | 400-th epoch | 600-th epoch | 800-th epoch | final epoch |
|---|---|---|---|---|
| CIFAR-10 | 60.8 | 83.3 | 97.8 | 99.3 |
| CIFAR-20 | 72.1 | 80.6 | 97.2 | 99.8 |

*Table 4.* Ablation study (in percent %) on CIFAR-10.

|  | NMI | ACC | ARI |
|---|---|---|---|
| BYOL | 72.3 | 83.3 | 68.0 |
| Only $L_s$ | 79.5 | 86.7 | 75.1 |
| LFSS w/o $L_I$ | 83.6 | 89.6 | 80.3 |
| LFSS w/o $L_C$ | 81.6 | 89.5 | 79.5 |
| LFSS w/o predecessor network | 83.5 | 90.7 | 81.5 |
| LFSS | **87.2** | **93.4** | **86.6** |

## 5.5. LFSS Benefits Unstable Samples

To further investigate the impact of LFSS, we evaluate the clustering accuracy of samples with different degrees of stability in Figure 4. We replicate the experimental setup in Observation 1, binning samples by stability and measuring the respective clustering accuracy. At the 1000-th epoch on CIFAR-10 and CIFAR-20, where LFSS has completed its training stage, it can be seen that LFSS (in red) improves the clustering accuracy of unstable samples compared to other methods. On CIFAR-10, the performance improvement of LFSS mainly comes from more accurate prediction of unstable samples. On CIFAR-20, LFSS enhances clustering performance for both unstable and stable samples. As higher clustering accuracy signifies more discriminative representations, these experimental results demonstrate that LFSS facilitates the learning of unstable samples.

Also, we focus on the top 10% most unstable samples at the 200-th epoch on CIFAR-10 and CIFAR-20, using the stability of the sample at exactly the 10-th percentile as the threshold for determining instability. In subsequent epochs of training, a significant proportion of these samples surpass this threshold and become stable. This indicates that our method can effectively reduce the number of unstable samples. The results are shown at Table 5.

## 5.6. Applying Sample Stability into Other Methods

The proposed LFSS is based on the BYOL framework (Grill et al., 2020). However, utilizing sample stability as a supervision signal is not limited to this framework. It can be extended to other methods. To better illustrate the generality of using sample stability as a supervision signal, we have extended LFSS to Moco (He et al., 2020) and SimCLR (Chen et al., 2020) by introducing $L_I$ and $L_C$ into their frameworks. For LFSS (MoCo) and LFSS (SimCLR), we adopt their respective network architectures and replace the $L_S$ in LFSS with their original loss functions. We train all models for 1,000 epoch with ResNet-18, using a batch size of 256. To achieve a fair comparison, we implement both the original methods and their corresponding LFSS versions using exactly the same experimental setup. For other experimental settings and hyper-parameter choices unique to LFSS, we keep them consistent with Section 5.2. The clustering performance is shown in Table 6. We can observe that LFSS exhibits performance improvements across various frameworks. In some experimental scenarios, such as Moco and BYOL on CIFAR-10, LFSS can bring about up

*Table 6.* Clustering performance (in percent %) of applying sample stability into various methods.

| Method | CIFAR-10 | | | CIFAR-20 | | |
|---|---|---|---|---|---|---|
|  | NMI | ACC | ARI | NMI | ACC | ARI |
| BYOL | 72.3 | 83.3 | 68.0 | 51.9 | 49.3 | 34.5 |
| LFSS (BYOL) | **87.2** | **93.4** | **86.6** | **59.9** | **58.7** | **43.5** |
| Moco | 50.5 | 57.7 | 40.2 | 34.8 | 36.9 | 19.3 |
| LFSS (Moco) | **56.8** | **67.6** | **49.0** | **39.3** | **40.9** | **22.0** |
| SimCLR | 73.9 | 83.1 | 69.0 | 47.2 | 46.8 | 31.2 |
| LFSS (SimCLR) | **79.2** | **88.4** | **74.4** | **48.2** | **47.5** | **31.4** |

to approximately a 10% performance improvement. These results substantiate the scalability of using sample stability as a supervision signal.

## 6. Conclusion

In this paper, we have presented a novel deep clustering method that harnesses insights from sample stability to enhance clustering outcomes. Through extensive experimentation, we established a strong correlation between sample stability, clustering accuracy, and network memorization. Notably, we revealed that unstable samples are more prone to misclustering, highlighting the utility of sample stability as a guiding signal in unsupervised clustering scenarios. Furthermore, we found that samples exhibiting prolonged instability pose challenges for network memorization, often characterized by complexity and uniqueness in their original features. Building on these findings, we proposed a novel LFSS method, which leverages sample stability at both the instance and cluster levels to advance deep clustering performance. Our method outperforms state-of-the-art approaches across various commonly employed datasets, demonstrating its efficacy. Through additional experiments, we delve into the factors underpinning our method's effectiveness and showcase its adaptability across diverse methods.

## Impact Statement

This paper presents work whose goal is to advance the field of deep clustering. There are many potential societal consequences of our work, none which we feel must be specifically highlighted here.

## Acknowledgments

This work was in part supported by the National Natural Science Foundation of China under Grant U24A20322 and 62422118, and in part supported by the Hong Kong UGC under grant UGC/FDS11/E02/22 and UGC/FDS11/E03/24. This research work is supported by the Big Data Computing Center of Southeast University.

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

# Appendix

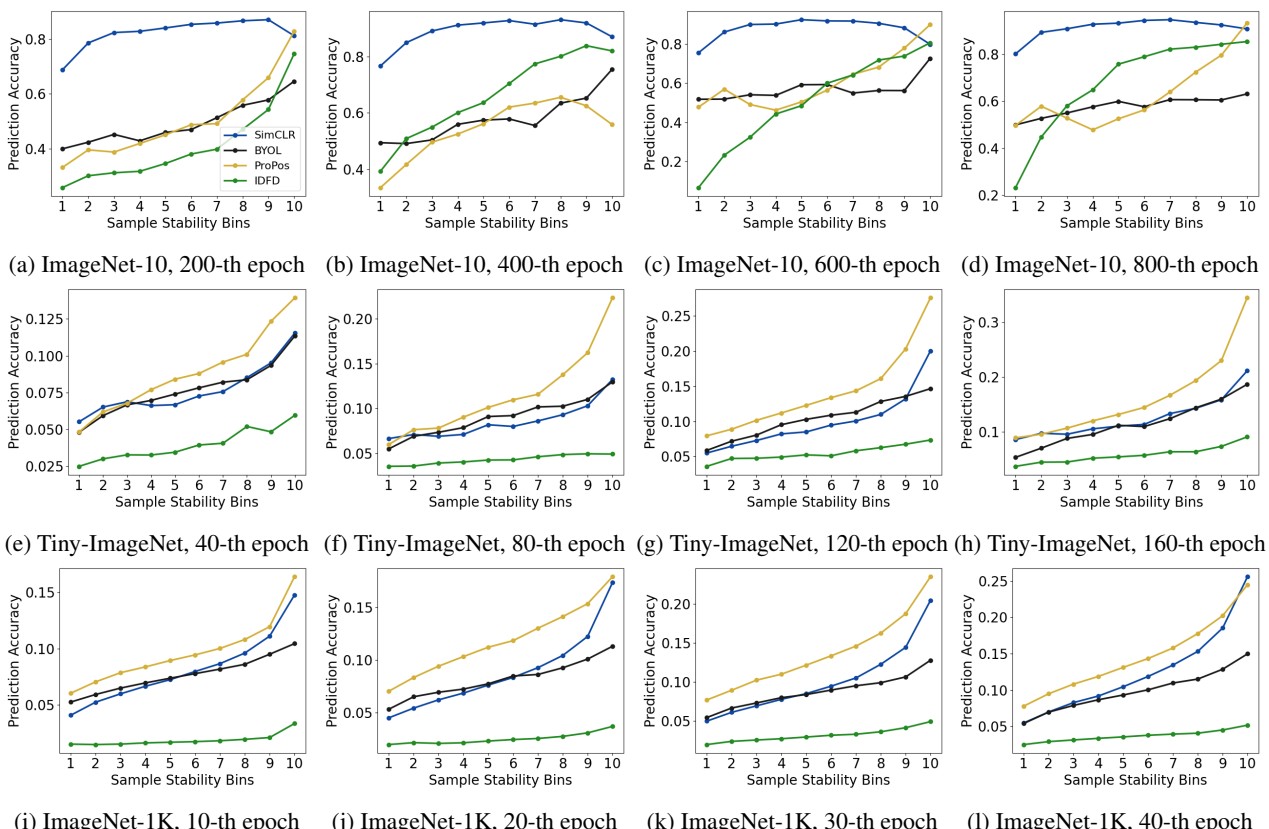

(a) ImageNet-10, 200-th epoch    (b) ImageNet-10, 400-th epoch    (c) ImageNet-10, 600-th epoch    (d) ImageNet-10, 800-th epoch

(e) Tiny-ImageNet, 40-th epoch    (f) Tiny-ImageNet, 80-th epoch    (g) Tiny-ImageNet, 120-th epoch (h) Tiny-ImageNet, 160-th epoch

(i) ImageNet-1K, 10-th epoch    (j) ImageNet-1K, 20-th epoch    (k) ImageNet-1K, 30-th epoch    (l) ImageNet-1K, 40-th epoch

*Figure 5.* Connection between sample stability and clustering prediction on ImageNet-10, Tiny-ImageNet and ImageNet-1K. The experiments indicate that across datasets of varying scales and at different training epochs, unstable samples are more likely to be predicted incorrectly in deep clustering. Four methods under various epochs: SimCLR (blue), BYOL (black), ProPos (yellow), IDFD (green). All subfigures share the same legend.

## A. More Experiments on Observation 1

Here, we provide further evidence for Observation 1. In addition to the CIFAR-10 and CIFAR-20 datasets mentioned in the main text, in Figure 5 we have also conducted experiments on ImageNet-10, Tiny-ImageNet, and ImageNet-1K. ImageNet-10 and Tiny-ImageNet are subsets of ImageNet-1K. The specifics can be seen in Table 2. These three datasets differ significantly in scale. Additionally, for different datasets, we present the training progress at various stages. In our experiments, we present the outcomes at the 200-th, 400-th, 600-th, and 800-th epochs on ImageNet-10, at the 40-th, 80-th, 120-th, and 160-th epochs on Tiny-ImageNet, and at the 10-th, 20-th, 30-th, and 40-th epochs on ImageNet-1K. In Figure 5, the results indicate that unstable samples are more likely to be clustered into wrong classes at various epochs across different methods and datasets. These experiments further corroborate Observation 1.

In all our experiments, we set the number of bins to 10, dividing all samples into 10 equally-sized groups based on their stability. We also present results with 5 and 20 bins on CIFAR-10 in Figure 6, to confirm that our findings are robust across different granularities of sample stability partitioning. These results strongly indicate a significant association between sample stability and clustering prediction.

We provide the training details of the observational experiments here to ensure reproducibility. We have trained models using four unsupervised methods, which are very influential in deep clustering, including SimCLR (Chen et al., 2020), Moco (He et al., 2020), ProPos (Huang et al., 2023) and IDFD (Tao et al., 2021). For all methods, we use ResNet-18 as the backbone on CIFAR-10, CIFAR-20 and ImageNet-10, ResNet-50 on Tiny-ImageNet and ImageNet-1K. The same experimental

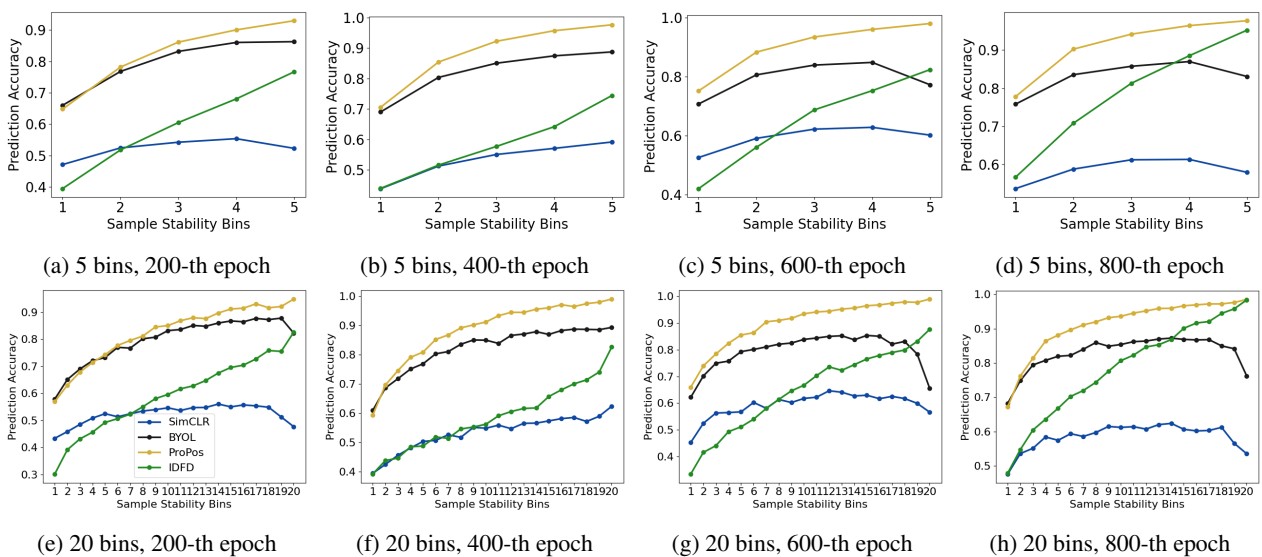

*Figure 6.* Observations on the stability of CIFAR-10 samples under partitioning granularity of different levels. It can be seen that across different partitions, unstable samples are more likely to be consistently predicted incorrectly. Four methods under various epochs: SimCLR (blue), BYOL (black), ProPos (yellow), IDFD (green). All subfigures share the same legend.

phenomena are observed across ResNets of different depths, further validating the generality of our observations. We adopt a batch size of 256 and train all models on an NVIDIA RTX 4090 GPU. We adopt the stochastic gradient descent (SGD) optimizer and the learning rate is set to 0.05 on CIFAR-10, CIFAR-20, Tiny-ImageNet and ImageNet-1K. We set learning rate to 0.005 on ImageNet-10 due to its smaller sample amount. Regarding the selection of method-specific hyper-parameters, we refer to their respective papers.

## B. More Experiments on Observation 2

We present the persistent instability of samples across various datasets in Table 7. As described in the main text, we identify a significant number of samples that exhibit long-term instability during the training process, far exceeding what would be expected by random selection. We selected four epochs on each dataset for statistical analysis, and these epochs are consistent with those used in the experiments described in Observation 1: we count the most unstable 10% of samples at the 200-th, 400-th, 600-th, 800-th epochs for CIFAR-20 and ImageNet-10, at the 40-th, 80-th, 120-th, 160-th epochs for Tiny-ImageNet, at the 10-th, 20-th, 30-th, 40-th for ImageNet-1K. Additionally, the probability of each sample being

*Table 7.* Recurrence counts of unstable samples on various datasets. In all datasets, there are certain samples that remain unstable over the long term.

| Dataset | Count | SimCLR | BYOL | ProPos | IDFD | Random |
|---|---|---|---|---|---|---|
| CIFAR-20 | Count = 1 | 11897 | 10734 | 10175 | 10915 | 17496 |
| | Count = 2 | 3870 | 4037 | 4109 | 4049 | 2916 |
| | Count = 3 | 1165 | 1364 | 1469 | 1273 | 216 |
| | Count = 4 | 217 | 275 | 300 | 292 | 6 |
| ImageNet-10 | Count = 1 | 2187 | 2753 | 1296 | 2868 | 3790 |
| | Count = 2 | 793 | 658 | 576 | 892 | 632 |
| | Count = 3 | 317 | 289 | 448 | 168 | 47 |
| | Count = 4 | 119 | 66 | 352 | 11 | 1 |
| Tiny-ImageNet | Count = 1 | 24109 | 20470 | 21453 | 17855 | 29160 |
| | Count = 2 | 6038 | 6182 | 6451 | 7026 | 4860 |
| | Count = 3 | 1129 | 1870 | 1586 | 2075 | 360 |
| | Count = 4 | 107 | 389 | 222 | 467 | 10 |
| ImageNet-1K | Count = 1 | 280545 | 269498 | 247905 | 233076 | 373587 |
| | Count = 2 | 81502 | 81466 | 83144 | 88438 | 62258 |
| | Count = 3 | 19289 | 21828 | 26250 | 26368 | 4612 |
| | Count = 4 | 2762 | 3638 | 4881 | 5852 | 128 |

*Table 8.* Comparisons with various methods on clustering performance (in percent %) on Tiny-ImageNet.

|     | DCCM | PICA | CC | GCC | MoCo | PCL | SimSiam | BYOL | PropPos | Ours |
|-----|------|------|------|------|------|------|---------|------|---------|------|
| NMI | 22.4 | 27.7 | 34.0 | 34.7 | 34.2 | 35.0 | 35.1 | 36.5 | 40.5 | **43.5** |
| ACC | 10.8 | 9.8 | 14.0 | 13.8 | 16.0 | 15.9 | 20.3 | 19.9 | 25.6 | **26.8** |
| ARI | 3.8 | 4.0 | 7.1 | 7.5 | 8.0 | 8.7 | 9.4 | 10.0 | 14.3 | **16.2** |

*Table 9.* Comparisons with various methods on clustering performance (in percent %) on ImageNet-1K.

|     | SimCLR | BYOL | IDFD | PropPos | Ours |
|-----|--------|------|------|---------|------|
| NMI | 40.6 | 40.6 | 24.6 | 43.2 | **45.4** |
| ACC | 15.2 | 11.5 | 3.45 | 14.8 | **15.8** |
| ARI | 7.81 | 55.9 | 1.06 | 7.19 | **8.58** |

randomly selected is 0.1, following a binomial distribution. Note that the results of the random selection are rounded to the nearest integers.

## C. Clustering on Large-scale Datasets

To demonstrate the effectiveness of LFSS on the large-scale dataset, we train the model on the Tiny-ImageNet and ImageNet-1K to evaluate its performance. Tiny-ImageNet contains 200 classes and 100,000 samples. Imagenet-1K contains 1,000 classes and about 1.2 million samples. They are generally challenging for deep clustering due to its relatively large number of classes. It is difficult for deep clustering methods to effectively learn discriminative representations that can distinguish between the various classes on such datasets.

On Tiny-ImageNet, following (Huang et al., 2023), we train the model with ResNet-18 for 1,000 epochs with a batch size of 256. For hyper-parameters, we set $\eta$ to 200, $\lambda$ to 0.1, $\sigma$ to 0.001 and $\delta$ to 0.1. As shown in Table 8, the proposed LFSS achieves superior performance compared to state-of-the-art algorithms. Specifically, we can observe that our method has significantly improved the NMI, ACC, and ARI by 3%, 1.2%, and 1.9% respectively. On ImageNet-1K, we implement four different methods (Chen et al., 2020; Grill et al., 2020; Tao et al., 2021; Huang et al., 2023) and report their performance in Table 9. In detail, we train all the models with ResNet-50 for 50 epochs, using a batch size of 256. For hyper-parameters in LFSS, we set $\eta$ to 50, $\lambda$ to 0.1, $\sigma$ to 0.001 and $\delta$ to 0.1. As shown in Table 9, LFSS achieves the best clustering performance. It improves ACC by 1%, NMI by 2.2% and ARI by 1.39%. The experimental results demonstrate the effectiveness of LFSS on the large-scale dataset.

## D. Training Specifics

Here, we provide the experimental details omitted from the main text to facilitate the reproduction of the proposed LFSS. We adopt the data augmentation strategies in (Chen et al., 2020). We set the dimension of output representations to 256. The temperature $\tau$ is set to 0.5 for all experiments. We adopt the stochastic gradient descent (SGD) optimizer, whose learning rate is 0.05, momentum is 0.9 and weight decay is 0.0005. We adopt a cosine annealing learning rate strategy to adjust the learning rate at each iteration. Following (Huang et al., 2023), we set the learning rate of the predictor head $h$ to be 10 times that of the rest of the network, to benefit model training. We set the minimum learning rate to 0 for all datasets, while specially setting it to 0.025 for ImageNet-Dogs. Following (Chen et al., 2020), we modify ResNet-18 when training on CIFAR-10 and CIFAR-20, due to their small image sizes. We remove the first max-pooling layer and use a 3x3 kernel in the first convolutional layer.

## E. Hyper-parameter Analysis

In this section, we analyze how multiple hyper-parameters influence model training. We evaluate models under different choices of hyper-parameters to observe their impact on the model. We employ the single-variable control method in hyper-parameter analysis and conduct experiments on CIFAR-10, CIFAR-20 and ImageNet-10. That is, when studying the effect of a particular hyper-parameter, we keep other hyper-parameters constant and consistent with Section 5.2.

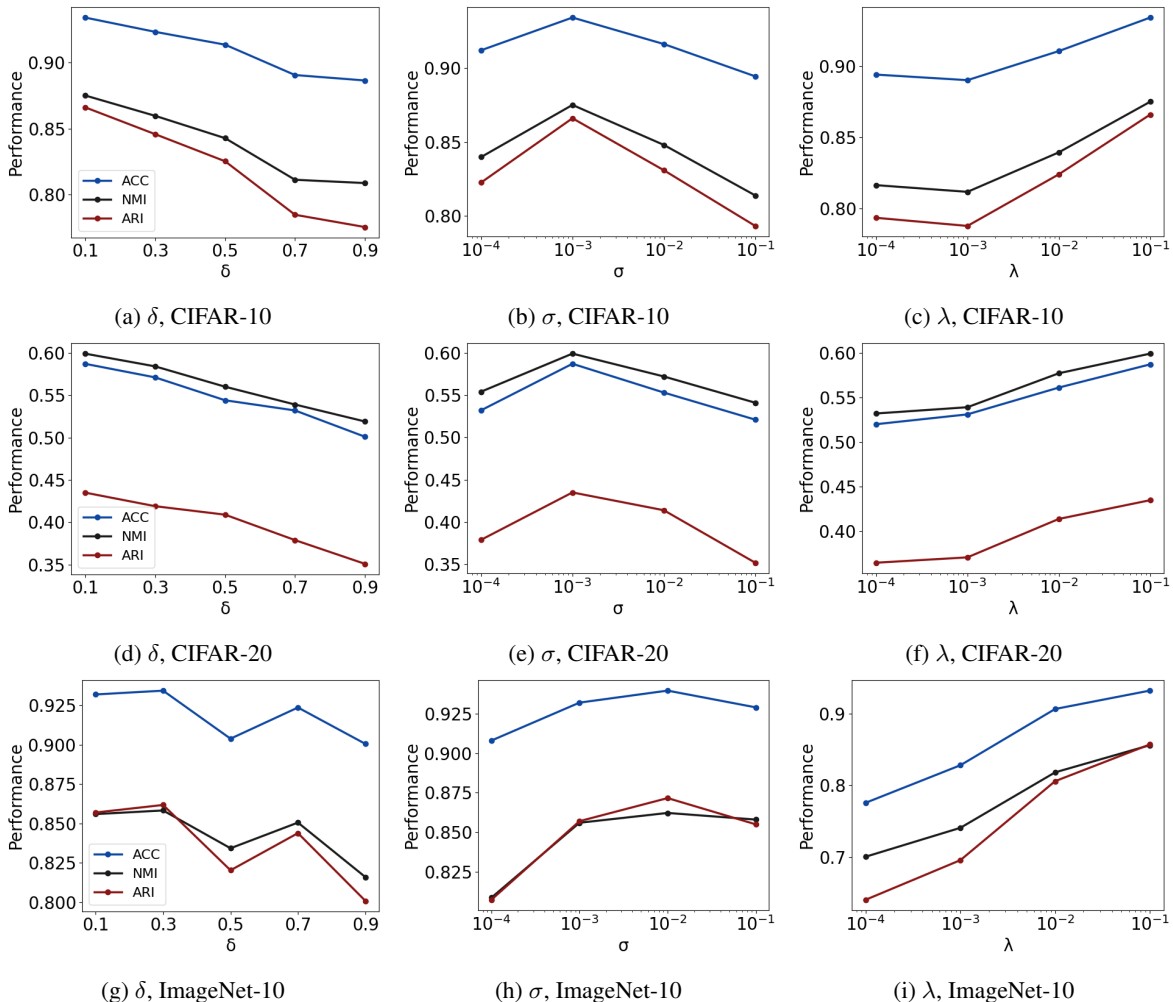

*Figure 7.* Hyper-parameter analysis of unstable ratio $\delta$, noise intensity $\sigma$ and balance parameter $\lambda$ on CIFAR-10, CIFAR-20 and ImageNet-10. All subfigures share the same legend.

**Unstable ratio $\delta$.** In cluster-level loss for LFSS, we employ cluster centers on contrastive learning and exclude a proportion of unstable samples from generating centers. We set the unstable ratio to 0.1, 0.3, 0.5, 0.7, 0.9 for investigating its effect. From Figures 7a, 7d and 7g, we can observe that as the unstable ratio $\delta$ increases from 0.1, all metrics show a declining trend. As the unstable ratio $\delta$ increases, the number of representations used to generate cluster centers decreases. When $\delta$ becomes too large, many accurate representations are also excluded, which can result in cluster centers that are not necessarily representative. In LFSS, we fix $\delta$ at 0.1.

**Noise intensity $\sigma$.** Noise generated from a normal distribution is introduced to the Eq. (9) for improving the robustness of model training. The intensity of this normally-distributed noise may have a notable impact on the experimental performance. To examine the impact of noise intensity $\sigma$ on clustering performance, we evaluate the models with $\sigma = 0.0001, 0.001, 0.01$, and 0.1. As shown in Figures 7b, 7e and 7h, we observed that as $\sigma$ varies, the clustering performance remains relatively stable. On different datasets, varying the noise level results in performance differences of about 5%, indicating that both excessively high and low noise intensity do not harm model training. In our experiments, we set $\sigma$ to 0.01 or 0.001.

**Balance parameter $\lambda$.** The parameter $\lambda$ balances the different terms in Eq. (10), determining the contribution of instance-level and cluster-level losses to model training. Self-distillation with noise $L_S$ differs from instance-level loss for LFSS $L_I$ and cluster-level loss for LFSS $L_C$ in mathematical forms, which results in different output values even when given the same input. We have investigated the impact of varying $\lambda$, including 0.1, 0.01, 0.001, and 0.0001, on the clustering

*Table 10.* Hyper-parameter analysis of $\eta$ on CIFAR-10, CIFAR-20 and ImageNet-10 (in percent %).

| Datasets | Metrics | $\eta = 0$ | $\eta = 50$ | $\eta = 200$ | $\eta = 500$ | $\eta = 800$ |
|---|---|---|---|---|---|---|
| | NMI | 78.1 | 79.2 | 87.2 | 85.7 | 86.4 |
| CIFAR-10 | ACC | 82.0 | 83.5 | 93.4 | 92.6 | 92.7 |
| | ARI | 71.1 | 71.8 | 86.6 | 84.9 | 85.5 |
| | NMI | 53.5 | 52.9 | 55.9 | 59.9 | 57.3 |
| CIFAR-20 | ACC | 50.9 | 48.1 | 53.4 | 58.7 | 56.2 |
| | ARI | 35.9 | 33.5 | 39.5 | 43.5 | 41.5 |
| | NMI | 64.3 | 71.1 | 85.6 | 83.1 | 83.7 |
| ImageNet-10 | ACC | 71.6 | 76.5 | 93.2 | 91.1 | 92.0 |
| | ARI | 54.6 | 62.8 | 85.7 | 81.9 | 83.1 |

*Table 11.* Significance Test of Various Methods

| | CIFAR-10 | | | CIFAR-20 | | | STL-10 | | | ImageNet-10 | | | ImageNet-Dogs | | |
|---|---|---|---|---|---|---|---|---|---|---|---|---|---|---|---|
| Method | NMI | ACC | ARI | NMI | ACC | ARI | NMI | ACC | ARI | NMI | ACC | ARI | NMI | ACC | ARI |
| IDFD | 71.4(0.8)* | 81.5(2.3)* | 66.6(1.1)* | 41.8(0.2)* | 41.1(0.2)* | 26.5(0.1)* | 62.3(0.1)* | 74.1(0.3)* | 55.0(0.2)* | 75.5(1.4)* | 86.2(2.7)* | 69.0(2.1)* | 50.2(1.2)* | 55.3(2.7)* | 36.9(1.0)* |
| ProPos | 85.6(0.0)* | 92.0(0.0)* | 84.1(0.0)* | 54.9(1.2)* | 51.8(1.6)* | 37.8(1.1)* | 72.1(1.3)* | 83.2(2.1)* | 70.4(1.6)* | 81.2(0.0)* | 89.0(0.0)* | 78.7(0.0)* | 54.8(2.6)* | 63.9(2.1)* | 45.2(2.2)* |
| CoNR | 78.1(1.2)* | 85.3(3.1)* | 71.1(1.6)* | 53.5(0.3)* | 49.9(0.4)* | 35.8(0.0)* | 70.2(0.2)* | 81.2(0.2)* | 69.2(0.4)* | 79.2(1.1)* | 87.7(3.1)* | 75.0(1.2)* | 52.7(1.3)* | 62.2(2.3)* | 42.1(1.1)* |
| DMICC | 72.2(0.2)* | 83.1(0.3)* | 67.5(0.3)* | 40.0(0.1)* | 41.8(0.2)* | 24.3(0.2)* | 64.1(0.0)* | 75.5(0.0)* | 60.5(0.0)* | 84.3(0.0)* | 92.5(0.0)* | 85.2(0.0)* | 55.1(0.1)* | 54.2(0.3)* | 39.8(0.1)* |
| BYOL | 72.3(0.4)* | 83.3(0.6)* | 68.0(0.8)* | 51.9(1.5)* | 49.3(2.2)* | 34.5(2.3)* | 68.0(0.2)* | 80.7(0.1)* | 63.5(0.2)* | 73.2(0.0)* | 82.1(0.0)* | 67.4(0.0)* | 51.8(1.1)* | 59.7(1.6)* | 41.3(1.6)* |
| LFSS (Ours) | **87.2(0.0)** | **93.4(0.0)** | **86.6(0.0)** | **59.9(0.0)** | **58.7(0.0)** | **43.5(0.0)** | **77.1(0.3)** | **86.1(0.3)** | **74.0(0.6)** | **85.6(0.0)** | **93.2(0.0)** | **85.7(0.0)** | **61.7(0.0)** | **69.1(0.0)** | **53.3(0.0)** |

performance. In Figures 7c, 7f and 7i, we can see that an overly small $\lambda$ can lead to performance degradation due to the insufficient contribution of these losses. In LFSS, we fix $\lambda$ at 0.1.

**Warmup epoch number** $\eta$. We set $\eta$ to 0, 50, 200, 500, and 800 to evaluate the impact of introducing cluster-level loss at different training stages. As shown in Table 10, early introduction ($\eta = 0$ or 50) reduces clustering performance due to generating cluster centers of low quality, while later introduction, when representative cluster centers are well established, achieves better final performance. In our experiments, we set $\eta$ to 200 or 500.

# F. Significance Test

The clustering performance we report on the reproduced methods are based on the average of 10 runs of K-means. In Table 11, we report the standard deviation for each method in the parentheses. We perform the t-test for significance at the 5% level to compare the results of the proposed method with those reproduced methods. * denotes rejection of the original hypothesis and the two results are significantly different. We can observe that the labels assigned by our method under K-means are very stable and our clustering performance is significantly better than those of other methods in all cases according to the t-test.

