# OpenReview forum: "Learning from Sample Stability for Deep Clustering"
_ICML.cc/2025/Conference — ICML 2025 poster_

### Official Review · Reviewer_vooh · 2025-03-02

**Overall Recommendation:** 3

**Summary:**

A deep clustering method based on the idea that unstable points, whose representations change a lot each epoch, are more likely to be inaccurately clustered. The main proposals are a loss function to encourage representation stability, and the exclusion of unstable points from training.

**Claims And Evidence:**

The ablation experiments are convincing, especially the demonstration that LFSS can be used as an add-on in existing methods. I have some questions about the main results.

The comparison methods in Table 3 are referred to as state-of-the-art, but I am not sure this is the case. Looking [here](https://paperswithcode.com/task/image-clustering) it appears that there are several methods that perform better than the ones you compare against, and also better than your method.

The results for ProPros reported in their paper are higher than you have in Table 3, and in fact higher than your method. Can you explain this difference?

If you are displaying results as reported as baselines in Huang et al. (2023), why not display all baseline results, or at least the best-performing ones?

**Essential References Not Discussed:**

A similar method is proposed in [1]. That method excludes samples that don't receive the same aligned cluster label across epochs. It differs from your method in that it takes the stability of cluster labels, rather than embeddings, but is also based on the idea that unstable points are less likely to be correctly clustered. Another paper with a similar idea is [2], which trains multiple models in parallel, and only trains on "confident" points that are clustered the same way in every model. I believe the first paper is relevant in the context of the claim that few existing methods have used instance-level stability for deep clustering. For the second, I leave it up to the authors as to whether it is sufficiently relevant.

[1] Mahon & Lukasiewicz, 2023, Efficient deep clustering of human activities and how to improve evaluation.
[2] Mahon & Lukasiewicz, 2021. Selective Pseudo-label Clustering.

**Experimental Designs Or Analyses:**

The results are mostly good. I have some questions about the scores for comparison methods, see above. I would also be interested to see whether LFSS actually reduces the number of unstable points.

**Methods And Evaluation Criteria:**

The method makes intuitive sense and appears to be implemented in a reasonable way.

**Other Comments Or Suggestions:**

line 357, LHS: "are representative approaches recently" -> "are recent representative approaches"?

**Other Strengths And Weaknesses:**

The presence of three different hyperparameters is a drawback, as they it be slightly cumbersome to choose appropriate values in a given application.

**Questions For Authors:**

are excluded points excluded from all three loss functions or just the cluster loss?

lines 253-255, RHS: why do you conduct k-means before excluding unstable points?

why is Figure 4. based on Epoch 600 instead of the final trained model?

**Relation To Broader Scientific Literature:**

Relation to scientific literature is adequate.

**Theoretical Claims:**

n/a

---

> ### Author Rebuttal · Authors · 2025-04-01
>
> > **On main results (Claims And Evidence)**
>
> We compared four state-of-the-art (SOTA), including IDFD, ProPos, CoNR, DMICC, the baseline BYOL by reproducing them, with the same experimental setup as ours, including backbone, batch size, number of epochs, etc. These experimental settings can significantly affect the model's performance and lead to unfair comparison if non-unified. For other representative methods, we directly adopted the results in ProPos. Below, we provide further clarification.
>
> About the compared SOTA: some existing methods surpass the performance of the SOTA we compared, as shown on the website you provided. However, to ensure a fair comparison, the selected SOTA are all methods based on representation learning for deep clustering. These methods focus on learning discriminative representations and then apply K-means to get the final clustering assignments, consistent with the approach used in LFSS. **Many of the higher-performing methods available on that website rely on leveraging the substantial knowledge embedded in pre-trained large foundation models**. Given that these methods do not learn representation from scratch and benefit significantly from their pre-trained foundations, comparing against them would be inherently unfair. Thus, we chose not to include these methods in our comparison, ensuring our comparisons remain equitable.
>
> About the reported results of ProPos: Due to the varying experimental settings of the SOTA methods, **we adopted a unified setup for fair comparison**. We used ResNet-18 instead of ProPos's original ResNet-34 and trained on a single GPU, resulting in slightly lower performance than their original results.
>
> Why not display all from results from ProPos：In fact, ProPos directly cited results from various papers. While many studies adopt this practice, it can lead to unfair experimental comparisons. For instance, a method trained for 3000 epochs may claim superior performance over one trained for only 300 epochs. Given limited computing resources, **we believe it is essential to at least reproduce the most relevant and competitive SOTA methods to validate performance fairly**.
>
> All the reproduced results we report are credible. We clearly indicate in the paper which methods are cited and which are reproduced, along with the experimental settings used for reproduction. We sincerely hope that these explanations will help you recognize the validity of our approach.
>
> > **Reducing unstable points (Experimental Analyses)**
>
> We focused on the top 10% most unstable samples at the 200th epoch on CIFAR-10 and CIFAR-20, using the stability of the sample at exactly the 10th percentile as the threshold for determining instability. In subsequent epochs of training, a significant proportion of these samples surpassed this threshold and became stable. This indicates that **our method can effectively reduce the number of unstable samples**. The results are at https://anonymous.4open.science/r/ICML25-0E61/4.1.png.
> > **Essential References Not Discussed**
>
> Both papers exclude unreliable pseudo-labels during training, due to inconsistent predictions across consecutive epochs and across multiple models respectively. In contrast, LFSS leverages sample stability at the representation level. It allows for training representations from scratch and can be embedded into multiple frameworks. We clarify that our contribution lies not only in the utilization of sample stability at the represention level or good experimental performance, **but more importantly, in uncovering the relationship among sample stability, clustering prediction and network memorization**. Meanwhile, the two papers you provided are relevant to our work, and we will cite and discuss them in the final version.
> > **About hyperparameters （Weakness）**
>
> Despite multiple parameters, LFSS is stable in parameter choice. Please refer to the first answer to Reviewer 9ozz for details. Thank you.
> > **On excluded points (Q1)**
>
> Only cluster loss, for more accurate centers. For other losses, we do not exclude them, allowing more samples to contribute to training.
> > **On the order of executing K-means (Q2)**
>
> The approach of excluding unstable samples first and then applying K-means is actually a similar process to performing K-means first and then excluding unstable samples. We trained LFSS with two methods on CIFAR-10 and recorded the clustering accuracy of the non-excluded samples at different epochs, to reflect the precision of the cluster centers obtained by two methods. The results are at https://anonymous.4open.science/r/ICML25-0E61/4.2.png. Two results are roughly similar, suggesting the two methods can achieve comparable outcomes.
> > **On Fig.4 (Q3)**
>
> Sorry for the confusion. To clarify, we presented results at the 600th epochs as an example mid-training. Similar effects are seen at final models in https://anonymous.4open.science/r/ICML25-0E61/4.3.png.
>
> Thank you for pointing out the typo and we will revise it.

---

> > ### Comment · Reviewer_vooh · 2025-04-04
> >
> > Thank you for the reply.
> >
> > The point about reproducing results is valid, and the additional experiments have done a good job of satisfying my concerns. I would definitely suggest including some of these results in the paper, particularly the change in the number of stable points and the final model's stability.
> >
> > I have raised my score to a 3.

---

> > > ### Author Response · Authors · 2025-04-04
> > >
> > > Thank you for your recognition of this work.
> > >
> > > We will revise the paper according to your suggestions. Specifically, we will cite and discuss the relevant works in Section 3, as suggested. We will include the experimental results on the change in the number of stable points and the benefits of LFSS for unstable samples in the final model in Section 5. Besides, we will rewrite Appendix E and provide a more detailed discussion on the sensitivity analysis of multiple parameters.
> > >
> > > We deeply appreciate the time and effort you have contributed to improving this work.

---

### Official Review · Reviewer_ZpFb · 2025-03-10

**Overall Recommendation:** 4

**Summary:**

This paper proposes a deep clustering method by identifying hard samples based on their stability during the training. By taking the sample stability into consideration, the proposed method improves instance-level representation learning and cluster-level grouping, leading to superior clustering results on five image datasets.

**Claims And Evidence:**

The proposed method is grounded on the observations of unstable samples. These observations are proven on different datasets with various methods, making this work technically sound. The utilization strategy for these unstable samples is also reasonable, with ablation studies demonstrating its effectiveness.

**Essential References Not Discussed:**

The authors are encouraged to include a recent deep clustering survey (A survey on deep clustering: from the prior perspective, Vicinagearth 2024), and a recent deep clustering method that also focuses on mining hard and valuable samples (Interactive Deep Clustering via Value Mining, NeurIPS 2024) in the related work section.

**Experimental Designs Or Analyses:**

The proposed method is evaluated on five classic and two large-scale image clustering datasets. The performance comparisons are fair to demonstrate the superiority of the method. Ablation studies and parameter analysis are also conducted to further interpret the effectiveness and robustness of the proposed method.

**Methods And Evaluation Criteria:**

The proposed method is technically sound, and the evaluations on five datasets are convincing enough to demonstrate the effectiveness of the method.

**Other Comments Or Suggestions:**

When referring to the three observations in the introduction section, the authors could add hyperlinks to help locate the details to improve readability.

**Other Strengths And Weaknesses:**

This paper reveals the strong correlation between samples' stability and their clustering accuracy. This finding could inspire future work in handling hard samples in deep clustering.

The proposed enhancement strategy using sample stability generalizes to different representation learning methods.

Applying k-means to compute the cluster centers could limit the scalability of the proposed method on large datasets.

**Questions For Authors:**

I expect the authors to respond to my previous concerns. In addition, while the proposed stability measure could help identify hard samples, how is this criterion different from commonly used confidence-based sample selection methods? For example, the authors may compare the intersection between hard samples selected according to different criteria and strategies.

Is the self-distillation with noise strategy a novel contribution of this work? It seems that augmenting with Gaussian noise is a commonly used trick. If this part is not novel, please correctly cite the corresponding works.

Minor: Is the place of the predictor head correct in Fig. 3?

**Relation To Broader Scientific Literature:**

This work might inspire researchers interested in developing clustering methods for other forms of scientific data.

**Theoretical Claims:**

This work has no theoretical claims.

---

> ### Author Rebuttal · Authors · 2025-04-01
>
> Thank you for your appreciation of this work. We highly value the insightful comments you have provided, and below we offer our responses.
>
> >　**Essential references not discussed**
>
> Thank you for providing the two recent related articles. After careful reading, we believe that these two papers are highly relevant to our work and should be cited in the introduction and related work sections in our work. The first article discusses the approach of solving deep clustering from a prior perspective, which is relevant to our discussion on constructing different supervisory signals based on prior knowledge. The second article innovatively proposes an unsupervised hard sample mining method and employs external user interaction to enhance clustering performance. We enhance clustering performance by utilizing sample stability, which can also be regarded as an indirect approach to hard sample mining. We will cite the two relevant paper in the final version.
>
> >　**The application of K-means is limited when it comes to large datasets (Weakness)**
>
> Indeed, applying K-means to obtain clustering assignments at every epoch can be very time-consuming, especially on large datasets. **However, the cluster-level loss for LFSS is only enabled after η epochs warmup, which means that the network can produce meaningful embeddings and does not necessarily require cluster assignments to be updated in every epoch**. We can reuse the clustering assignments obtained from a single K-means run across multiple epochs. We evaluate model performance under different cluster assignment update frequencies on CIFAR-10 in https://anonymous.4open.science/r/ICML25-0E61/3.1.png. Although the strategy of updating at every epoch achieves the best performance, the performance drop is marginal when the epoch interval is set to 10, 50, or 100. This allows us to significantly reduce training time. Therefore, we can achieve near-optimal performance while saving time by reducing the execution frequency of K-means.
>
> > **Hyperlink to observations (Suggestion)**
>
> We will add hyperlinks in Introduction section that point to specific observations to enhance readability in the final version.
>
> > **Comparison with confidence-based sample selection (Question 1)**
>
> The distinction between these two approaches lies in their application scenarios. The confidence-based sample selection method typically requires pre-training a representation model first, then training the clustering head with high-confidence pseudo labels to improve performance. This approach's success hinges on the model's ability to generate good representations and pseudo labels. In contrast, our method is a training-from-scratch approach that can be used for representation model training without the need for pre-training.
>
> We apply the self-labeling method in SCAN [1] on the final model of ours on CIFAR-10, using a threshold of 0.99 to filter out low-confidence pseudo-labels. Some of the low-confidence samples excluded by this method overlap with the unstable samples we selected from the final model. We count the number of these samples and provided their accuracy under K-means clustering as below:
>
> | Sample Type    |Accuracy|Quantity|
> |----------------|--------|--------|
> |Unstable Samples| 0.791  | 5998   |
> |Low Confidence  | 0.934  | 12008  |
> |Intersection    | 0.809  | 1273   |
>
> The accuracy of the selected unstable samples is lower, while the accuracy of the low-confidence samples is close to the overall accuracy of the global dataset. **This demonstrates that the sample stability-based method can identify misclustered samples more effectively than confidence-based approaches**.
>
> > **Question on self-distillation with noise strategy (Question 2)**
>
> self-distillation with noise strategy is not a novel contribution in this work and it is a commmonly used trick. This reparameterization trick is oriented in VAE [2] and we should cite it in our paper.
>
> > **Minor in Fig.3**
>
>  We sincerely thank you for pointing out this typo. The framework of LFSS is built upon BYOL, where the predictor head is connected after the online network. We will revise this in the final version.
>
> Heartfelt thanks for your efforts in this reivew.
>
> [1] Scan: Learning to classify images without labels, ECCV, 2020.
>
> [2] Auto-encoding variational bayes, ICLR, 2014.

---

> > ### Comment · Reviewer_ZpFb · 2025-04-02
> >
> > Thanks for the responses. I especially like the additional results on the comparison with confidence-based sample selection. My concerns have been addressed and I would like to raise my score to accept.

---

> > > ### Author Response · Authors · 2025-04-02
> > >
> > > Thank you very much for your careful review. We are grateful for raising our score. We are pleased to address your concerns and thank you for the time and effort you have dedicated to this work.

---

### Official Review · Reviewer_9ozz · 2025-03-14

**Overall Recommendation:** 3

**Summary:**

This article introduces LFSS, a novel deep clustering method that leverages sample stability, which is measured as the cosine similarity between representations across consecutive training epochs as a supervisory signal. The authors motivate the approach by showing that samples with unstable representations tend to be misclustered and are harder for networks to memorize. Based on extensive empirical observations across various datasets and multiple baselines, the paper proposes two key contributions: an instance-level loss that directly penalizes representation instability and a cluster-level loss that improves the quality of cluster centers by excluding the most unstable samples. The method is integrated into a self-distillation framework with noise, and experiments show significant improvements over baseline unsupervised methods and state-of-the-art deep clustering techniques. Necessary experiments were conducted to validate the claim of the paper.

**Claims And Evidence:**

The paper is clear about its motivation and claims with a good presentation, sufficient significance, quality, and originality.

**Essential References Not Discussed:**

N/A

**Experimental Designs Or Analyses:**

Necessary experiments were conducted to validate the claim of the paper.

**Methods And Evaluation Criteria:**

The proposed methods and evaluation criteria make sense for the claim of the paper.

**Other Comments Or Suggestions:**

1. Consider adding a more detailed analysis or visualization of the hyperparameter sensitivity.
2. Discuss the computational cost more explicitly, including any trade-offs in terms of training time or memory requirements.

**Other Strengths And Weaknesses:**

**Strengths:**
- The paper introduces the interesting concept of “sample stability” as a proxy for training progress and memorization difficulty in unsupervised learning. This perspective is both intuitive and well motivated.
- The experimental evaluation is thorough. The authors provide extensive results across a range of datasets and compare against multiple competitive baselines. Ablation studies further isolate the contributions of each component (instance-level loss, cluster-level loss, and noise-based self-distillation).

**Weaknesses:**
- While the method shows improved performance, it introduces several additional hyperparameters (e.g., the unstable ratio δ, warm-up epoch number η, noise intensity σ). A more detailed discussion of the sensitivity to these hyperparameters, or guidelines for tuning them across different datasets, is needed.
- The use of multiple network components (online, target, and predecessor networks) might introduce additional computational cost. The paper should provide a clearer discussion on the computational efficiency or training time compared to baseline methods.

**Questions For Authors:**

1. How robust is the LFSS framework when applied to datasets with significantly different characteristics (e.g., highly imbalanced data or non-image data)?
2. How does the additional computational overhead of maintaining a predecessor network compare with the performance benefits, especially in large-scale scenarios?
3. What is the computational complexity of LFSS?

**Relation To Broader Scientific Literature:**

The article tries to formulate a novel deep clustering algorithm which is significant in the deep clusering field.

**Theoretical Claims:**

The paper is clear about its theoretical claims.

---

> ### Author Rebuttal · Authors · 2025-04-01
>
> Thank you for your important questions. Below is our response:
> > **Hyperparameter Sensitivity (Weakness 1)**
>
> Actually, we provided an analysis of the sensitivity of four hyperparameters on CIFAR-10 in Appendix E. Here, we further provide a related analysis on three datasets: CIFAR-10, CIFAR-20, and ImageNet-10, along with a guideline to assist readers in applying the method to other datasets. The results are at https://anonymous.4open.science/r/ICML25-0E61/2.1.png for unstable ratio δ, noise intensity σ, balance parameter λ and https://anonymous.4open.science/r/ICML25-0E61/2.2.png for warmup epoch number η.
>
> **Unstable ratio δ**. Typically, a smaller δ is more effective, while a larger δ may exclude many meaningful representations from contributing to the clustering centers, thereby reducing model performance. **In LFSS, we fix δ at 0.1**.
>
> **Noise intensity σ**. Adding noise improves the model's robustness. On different datasets, varying the noise level results in performance differences of about 5%, indicating that both excessively high and low noise intensity do not harm model training. In our experiments, we set σ to 0.01 or 0.001.
>
> **Balance parameter λ**. The parameter λ balances the different terms in Eq. (10), determining the contribution of instance-level and cluster-level losses to model training. An overly small λ can lead to performance degradation due to the insufficient contribution of these losses. **In LFSS, we fix λ at 0.1**.
>
> **Warmup epoch number η**. We set η to 0, 50, 200, 500, and 800 to evaluate the impact of introducing cluster-level loss at different training stages. Early introduction (η = 0 or 50) reduces clustering performance due to generating cluster centers of low quality, while later introduction, when representative cluster centers are well established, achieves better final performance. In our experiments, we set η to 200 or 500.
>
> > **LFSS on imbalance datasets (Question 1)**
>
> To evaluate the robustness of LFSS on imbalanced datasets, we conducted experiments on CIFAR-10, CIFAR-20, and STL-10, using an imbalance ratio of 10. We adoped ResNet-18 with a batch size of 256 and trained for 1000 epoch. Besides the three metrics in the paper, we also used **CAA (class-averaged accuracy)** for evaluating performance on imbalanced datasets. The results are at https://anonymous.4open.science/r/ICML25-0E61/2.3.png. **Compared with state-of-the-art methods, e.g., IDFD, CoNR, ProPos, and DMICC, our method achieves the best performance on all three datasets on all four metrics with the maximum improvement over the second-best method is 7% on ACC**. Although none of the methods specifically address the handling of class imbalance, our approach demonstrates greater robustness on imbalanced datasets compared to other methods.
>
> > **LFSS on non-image data (Question 1)**
>
> Consistent with our comparison methods, we use image datasets to validate the effectiveness of our method. But our method is not limited to image datasets. As suggested, we further evaluated our method on two text datasets GoogleNews-T and GoogleNews-S. We replaced the ResNet-18 in LFSS with an MLP. We choose distilbert as the backbone to extract features from original texts.  We compared LFSS with text clustering methods such as BoW, TF-IDF and HAC-SD. We use the same experimental setup as ours to reproduce IDFD and ProPos. The results are at https://anonymous.4open.science/r/ICML25-0E61/2.4.png. LFSS shows **improvements of about 2-3% on all metrics compared with the second best**. Despite not specifically considering the data characteristics and design schemes for text clustering, our method still managed to deliver highly competitive experimental performance. This underscores the superiority of our approach and its robustness across different types of data.
>
> > **Computional cost (Weakness 2, Question 2)**
>
> In fact, adding a predecessor network into BYOL framework does **not introduce significant computational overhead, as this network is updated by directly copying the weights from the previous epoch, without requiring gradient computation or backpropagation**. We compared the runtime (**minute**) and memory usage (**MB**) for three methods based on the BYOL framework: BYOL, ProPos, and LFSS, as follows:
> |      | BYOL|Propos| LFSS  |
> |------|-----|------|-------|
> |Time  |327.8|416.8 | 456.9 |
> |Memory|4013 | 4137 | 4149  |
>
> It can be seen that LFSS's computational resource usage is slightly higher than other methods but remains within an acceptable range, all within the same order of magnitude.
>
> > **Computional complexity of LFSS (Question 3)**
>
> The computional complexity of LFSS loss is $O(N^2*d)$, where N is the batch size and d is the embedding dimension. We do not consider the computional complexity of the backbone or cluster assignment, as they are independent of our method.
>
> We hope the above response is satisfactory, and we thank you for the time and effort you devote to this review.

---

### Official Review · Reviewer_2B7a · 2025-03-15

**Overall Recommendation:** 4

**Summary:**

This work introduces a novel sample stability, which is strongly tied to misprediction and memorization difficulty. By leveraging stability as a supervision signal, the proposed LFSS method outperforms state-of-the-art approaches on multiple benchmarks.

## update after rebuttal

I read the rebuttal, which addressed my questions. I keep the score.

**Claims And Evidence:**

NA

**Essential References Not Discussed:**

NA

**Experimental Designs Or Analyses:**

NA

**Methods And Evaluation Criteria:**

NA

**Other Comments Or Suggestions:**

NA

**Other Strengths And Weaknesses:**

### Strengths
* The paper is clearly structured, and its motivation is explained in a way that is easy to understand.
* The experiments are extensive.
* The method achieves excellent performance on various benchmarks.

### Weaknesses
* By definition, Sample Stability is likely to be lower at the beginning of training (when representations are still evolving) and higher toward the end of training (when the model stabilizes). The paper observes that high stability corresponds to higher accuracy. A more detailed visualization (e.g., a histogram) showing how Sample Stability changes over the course of training would strengthen the empirical insights.

* The choice to update the final representation with the latest epoch only seems somewhat ad hoc. It would be informative to investigate updating at larger intervals (e.g., every few epochs) for both the representation update and the calculation of Sample Stability, to test whether this yields more reliable estimates or improved performance.

* Figure 2 suggests that samples with lower Sample Stability tend to be harder examples. One question is whether applying a hard example mining strategy could further boost performance—e.g., by giving these challenging samples additional training updates or specialized handling.

* It would be helpful to visualize how different types of samples (e.g., high vs. low stability, easy vs. hard) are distributed in a latent space via scatter plots or other methods. Such a visualization might reveal meaningful structure and give deeper insights into how and why certain samples remain unstable.

**Questions For Authors:**

NA

**Relation To Broader Scientific Literature:**

NA

**Theoretical Claims:**

NA

---

> ### Author Rebuttal · Authors · 2025-04-01
>
> Thank you for your careful review and constructive comments. Below is our response:
>
> > **Changes in sample stability during training (Weakness 1)**
>
> Thank you for your valuable advice. We conducted experiments to investigate the changes in sample stability as training progresses. The experiments were performed on SimCLR, BYOL, ProPos, and IDFD frameworks, consistent with the main text. We also tested the proposed LFSS under the same experimental setup. Following the Observation 1 in the main text, we divided the samples into ten bins based on their stability, ranging from lowest to highest. On the CIFAR-10 dataset, we calculated the mean sample stability of each bin at 200, 400, 600, and 800 epochs for each methods. The results are visualized in https://anonymous.4open.science/r/ICML25-0E61/1.1.png. **Generally, as training progresses, sample stability gradually increases. This phenomenon becomes more pronounced once the model training has stabilized after 400 epochs**.
>
> Also, the sample stability values differ among various methods due to their differing learning strategies. Methods like BYOL, ProPos, and LFSS, which use Exponential Moving Average (EMA) to update the target network, tend to have smoother network changes and thus exhibit overall higher sample stability compared to IDFD and SimCLR, which do not utilize EMA. However, as Observation 1 demonstrates, the relatively unstable samples in these methods still exhibit lower accuracy compared to the relatively more stable ones, confirming the applicability and robustness of our findings across methods that differ significantly in their characteristics.
>
> > **Experiments on larger intervals (Weakness 2)**
>
> Thank you for your deep thoughts on this work. Following on your suggestion, we conducted experiments on CIFAR-10 and ImageNet-10 to verify whether a larger interval would lead to performance improvement. We chose intervals of the smaller values 5, 10, and the larger value 100 to ensure the thoroughness of the experiments. The results are below：
>
> CIFAR-10:
>
> | Interval  | NMI  | ACC  | ARI  |
> |-----------|------|------|------|
> | 1 epoch   | **87.2** | **93.4** | **86.6** |
> | 5 epochs  | 85.8 | 92.2 | 84.4 |
> | 10 epochs | 84.4 | 91.3 | 82.7 |
> | 100 epochs| 80.4 | 88.5 | 77.7 |
>
> ImageNet-10:
>
> | Interval   | NMI  | ACC  | ARI  |
> |------------|------|------|------|
> | 1 epoch    | **85.6** | **93.2** | 85.7 |
> | 5 epochs   | 84.1 | 91.8 | 84.8 |
> | 10 epochs  | 85.5 | 92.5 | **86.1** |
> | 100 epochs | 82.0 | 89.1 | 81.2 |
> When the interval is set to 5 or 10 epochs, the clustering performance remains strong, albeit with a slight decline. However, when the interval increases to 100 epochs, the drop in model performance becomes more pronounced. We believe that when the interval is 1 epoch or a few epochs, sample stability can effectively reflect the training quality of the samples; that is, hard samples tend to be unstable. In contrast, with larger intervals, the gradual optimization of sample features during training also becomes a contributing factor influencing sample stability.
>
> > **Advice on hard sample strategy (Weakness 3)**
>
> Since unstable samples can be considered hard samples that are difficult to correctly identify, **LFSS can also be viewed as a method for improving performance in hard sample mining**. Particularly, in our cluster-level loss (Eq. (8)), we exclude unstable samples (hard samples) to ensure more accurate cluster centers for contrastive learning. Without this loss, performance on various metrics for CIFAR-10 would decrease by 4-7%, as described in ablation study. We look forward to further addressing unsupervised hard sample mining from the perspective of sample stability. We are grateful for your kind advice.
> > **Visualization on unstable and stable samples (Weakness 4)**
>
> We sincerely appreciate your suggestions for improving this work. We have visualized the distribution of embeddings for both stable and unstable samples in https://anonymous.4open.science/r/ICML25-0E61/1.2.png. We perform t-SNE on final embeddings on CIFAR-10. We mark top 10% stable samples in yellow,  top 10% unstable samples in purple and other samples in green. Even though the number of stable and unstable samples selected is similar, there appear to be more purple dots (unstable ones) on the figure intuitively. This indicates that **the distribution of unstable samples is more dispersed, while the distribution of stable samples is more concentrated, even resulting in many overlaps**. This suggests that stable samples tend to be representative samples with similar properties and characteristics. Meanwhile, we also observe that **isolated samples in the gaps between clusters often belong to the unstable ones, which highlights their atypical features**.
>
> We are very grateful for your sincere suggestions in improving our work as well as for your recognition of our effort. Thank you.

---

### Decision · Program_Chairs · 2025-05-01

**Decision:**

Accept (poster)

**Comment:**

Regarding this paper, all four reviews are positive. According to the reviews, the strengths of the work are as follows:
* The motivation is clear, and the idea of "sample stability" for clustering is interesting.
* The experimental evaluation is thorough, and the proposed algorithm outperformed the competitors in most cases.

I suggest the authors include the results and discussion produced during the rebuttal and discussion periods in the revised paper.